# Sexual dysfunction among men with diabetes mellitus attending chronic out-patient department at the three hospitals of Northwest Amhara region, Ethiopia: Prevalence and associated factors

**Eskedar Getie Mekonnen** ***, Hedija Yenus Yeshita, Alehegn Bishaw Geremew**

Department of Reproductive and Child Health, Institute of Public Health, College of Medicine and Health Sciences, University of Gondar, Gondar, Ethiopia

* eskedargetie18@gmail.com

## Abstract

### Background

Sexual dysfunction is the commonest reproductive health problem observed among men with diabetes mellitus affecting their quality of life. Previous studies conducted in this area were concentrated on the specific domains of sexual dysfunction, and factors were not well-addressed. Therefore, this study was aimed to determine the prevalence of all forms of sexual dysfunction and to identify its associated factors among diabetic men patients attending at the three hospitals of the Amhara region, Ethiopia.

### Method

An institutional-based cross-sectional study was conducted involving 462 men diabetic patients at the three hospitals of the northwest Amhara region. A systemic random sampling technique was employed. A face-to-face interviewer-administered change in the sexual functioning questionnaire was used to collect the required data from the 20th of February to the 15th of April 2020. The binary logistic regression was employed and a multivariable logistic regressions model was used to control the effect of confounders. Variables that had an independent correlation with the sexual dysfunction were identified based on a p-value$\leq$ 0.05. Likewise, the direction and strength of association were interpreted using Adjusted Odds Ratio (AOR) with its corresponding 95% CI.

### Results

The prevalence of sexual dysfunction was found to be 69.5% (95%CI: (65.1–73.9)). The magnitude of sexual dysfunction was prevalently observed among participants who were older (> 50 years) (AOR = 8.7, 95%CI: (3.3–23.1)). Likewise, the odds of sexual dysfunction was significantly higher among men who have lived with diabetes for a longer duration (AOR = 10.8, 95%CI: (5.3–21.9)), with poor metabolic control (AOR = 3.57, 95%CI:

**Funding:** The university of Gondar has funded the study but has no role in the study design data collection and analysis, decision to publish or preparation of the manuscript.

**Competing interests:** The authors have declared no competing interests exist.

**Abbreviations:** AOR, Adjusted Odd Ratio; BMI, Body Mass Index; BPH, Benign Prostatic Hyperplasia; COR, Crude Odd Ratio; CSFQ, Change in Sexual Function Questionnaire; CSI, Couple Satisfaction Index; DM, Diabetes Mellitus; DSEMS, Daily Stressful Event Measurement Scale; ED, Erectile Dysfunction; FBS, Fasting Blood Sugar; HTN, Hypertension; IIEF, International Index of Erectile Function; OPD, Outpatient Department; PE, Premature Ejaculation; RE, Retrograde Ejaculation; SD, Sexual Dysfunction; WHO, World Health Organization.

(1.81–7.05)), with comorbid illnesses (AOR = 5.07, 95%CI: (2.16–11.9)), and diabetic-related complications (AOR = 3.01, 95%CI: 1.31–6.92). On the other hand, participants who were physically active (AOR = 0.41, 95%CI: (0.12–0.7)) and satisfied with their relationship (AOR = 0.15, 95%CI: (0.03–0.7)) showed a lesser risk of experiencing sexual dysfunction.

## Conclusion

Well over two-thirds of men with diabetes mellitus have experienced sexual dysfunction, implying a public health pressing problem. Older age, lack of physical activity, living longer duration with diabetes, having diabetic complications, experiencing co-morbid illnesses, being unsatisfied with couple relationship, and poor metabolic control increased the risk of developing SD. Therefore, promoting physical exercise, preventing co-morbid illnesses, and couples counseling to build up a good couple relationship are recommended to promote the sexual and reproductive health of men with diabetes.

## Introduction

Sexual dysfunction (SD) is a multi-factorial and heterogeneous group of disorder which may take different forms mainly characterized by clinically significant disturbances in the people's ability to respond sexually, or to experience sexual pleasure [1–3]. SD might occur in both sexes but the problem in men tends to be more associated with physical health including chronic disease and aging [4–6]. SD in men is categorized based on the sexual response cycle that includes hypoactive sexual desire, arousal disorder (erectile dysfunction(ED)), and orgasm disorder (premature, retrograde ejaculation, and anorgasmia) [7, 8]. Patients with chronic disease(s) are more susceptible to developing SD related to physiological disruption, drugs side-effect, emotional disturbance, or the combination of those factors [6]. The mechanism by which diabetes results in SD is multifaceted that includes psychogenic, hemodynamic, neurogenic, hormonal, and atrophy of smooth muscle within the corpus cavernous bodies [6, 9, 10].

There has been a global increase in male sexual disorders [4]. The magnitude of sexual problems was higher in East and Southeast Asia than in other regions of the world in which a bit lower than a third (31%) of men aged between 40 and 80 declared to experience SD. In East Asia, around 29.1 and 27.1% of victims claimed to have early ejaculation and ED forms of SD, respectively [4, 11, 12]. ED is estimated to affect 152 million men worldwide [13]. In America, the prevalence of ED among men with diabetes ranges from 35 to 75% and it occurs at an earlier age [14]. Further, 12 million men are estimated to be affected by ED in Africa. In Nigeria in particular, more than three out of every 10 men have suffered from ED or another form of SD [2, 4, 15]. The magnitude of SD among men with DM varies across different settings that range from 51–85.5% [10, 16–19]; in Ethiopia, 69.9–85.5% of diabetic men were indicated to have ED [20, 21].

SD is ascribed to various modifiable and non-modifiable factors. Age is one of the commonest risk factors for an increased incidence of different SD domains and the problem is frequently observed among aged men with DM [14]. Studies evidenced that the age-adjusted risk of SD (ED) was doubled in diabetic men compared with those individuals without diabetes [22]. Likewise, comorbid illnesses, for instance, hypertension increases the risk of developing SD; an estimated 40 to 80% of diabetic patients with HTN were reported to have SD associated with the effect of the illness itself, the drug side effect, and the psychological impact of chronic

diseases [23–25]. Moreover, patients with micro and macrovascular diabetic complications are also at higher risk of SD [10, 26].

Although SD is a frequently observed complication of diabetes, studies are limited in many countries, including Ethiopia. In addition, studies conducted earlier had only focused on specific domains of SD, and thus, the exact burden would not be highlighted. This study was, therefore, conducted to assess the prevalence of SD among diabetic men, and to identify factors associated with SD. The findings of the study will positively influence the local decision-makers to mitigate the problem through working on the identified responsible factors that contribute to SD.

## Materials and methods

### Study setting

The study was conducted from the 20th of February to the 15th of April, 2020. Participants were recruited from the chronic outpatient department of Felege Hiwote comprehensive and specialized hospital (FHCSH), Debre Markos referral hospital, and Debre Tabor general hospital. The chronic out-patient department is the one among others where the number of diabetic patients account for the largest proportion of the chronic out-patient visit.

### Study design

An institutional-based cross-sectional study was employed among men with DM attending at the three hospitals of the northwest Amhara region.

### Sample size and sampling procedure

The required sample size was determined using the single population proportion formula considering the following statistical assumptions: prevalence of SD among men with diabetes mellitus as 65% [19], 4.5% of margin of error, a standard Z score of 1.96 corresponding to 95CI, and 10% non-response rate. Finally, the sample size was computed as:

$$n = \frac{Z_{\frac{\alpha}{2}}^2 \times p(1-p)}{d^2}; \quad n = \left[(1.96)^2 * 0.65(1-0.65)\right]/(0.0451)^2 = 420$$

After adding 10% none response rate the sample size was **462**.

The sample size was proportionally allocated to each hospital considering the monthly patient flow. Participants were approached in every other two men through systemic random sampling technique.

### Study population

Men with diabetes who came to the chronic OPD for monthly follow-up during the data collection period were invited to participate in the study. Screening was done to identify and recruit study participants who already have started sex. Victims who were disoriented, unable to communicate, and those who were currently sexually inactive due to different reasons (separated from partner and men who were catheterized) were excluded.

### Variables

**Dependent variable.** Sexual dysfunction (Yes/No)

**Independent variables.** *Socio-demographic factors*. Age, marital status, occupational status, educational status, etc.

*Medical conditions*. Comorbid illness and diabetes related factors

*Structural factors*. Benign prostatic hyperplasia, iatrogenic pelvic injuries, and pelvic radiation

*Behavioral and lifestyle factors*. Alcohol, smoking, body mass index, and physical activity

*Psychosocial factor*. Quality of relationship and stressful life event.

## Operational definitions

**Sexual dysfunction.** Explained by a total score below the cutoff points (47) from 70 for all 14-items of change in the sexual functioning questioner (CSFQ) [27].

**Sexual dissatisfaction.** Scoring less than 5 from CSFQ item 14 [27].

**Sexual desire disorder.** Scoring less than 20 from the sum of CSFQ-14- (items 2 through 6) [27].

**Arousal/Excitement dysfunction.** Explained by a score less than 14 from the sum of CSFQ-14- (items 7 through 9) [27].

**Anorgasmia.** Explained by score less than 14 from the sum of CSFQ-14- (items-11 through 13) [27].

**Sexual pain disorder.** Explained by score less than 5 from the CSFQ −14- (item ten) [27].

**Couple satisfaction status.** Was explained as satisfied if participants scored above 20 from the total of couple satisfaction index (CSI) [28].

**Stressful life event.** Was explained if respondents experienced at least one of the listed ten items from the daily stressful event measurement scale (DSEMS) in the past 6 months of the survey.

**Comorbid illness.** Existence of additional chronic illnesses, including hypertension, cardiac disease, dyslipidemia, psychosis, renal disease, HIV, cancer, asthma, and multiple sclerosis.

**Diabetic complications.** The existence of macrovascular diabetic complications, microvascular diabetic complications (retinopathy, neuropathy, and nephropathy), and diabetic foot ulcer.

**Poor glycemic control.** Current fasting blood glucose level greater than 130mg/dl or most recent HgA1c >9.0% reflecting poor glycemic control [29].

**Alcoholic.** The daily alcohol amount that respondents consume was calculated considering the average alcohol percent (%/ml) of each drink multiplied by the volume (ml) of the drink and volumetric mass density (which is 0.8g/ml). Accordingly, participants were explained to be alcoholic if they drink more than 12g/ethanol of alcohol per day in the past six months of the survey [30].

**Nutritional status.** Underweight: BMI<18.5kg/m$^2$; normal: 18.5–24.9kg/m$^2$; overweight: 25–29.9 kg/m$^2$; and obese: BMI > 30 kg/m$^2$.

**Smoker.** A respondent who smokes $\geq$ 12 cigarettes per day in the past six months of the survey [1].

## Data collection tool, procedure, and measurement

The data were collected through a face-to-face interviewer-administered questionnaire. Changes in Sexual Functioning Questionnaire (CSFQ-14) adapted from reliability and construct validity of the changes in sexual functioning questionnaire short-form (CSFQ-14) [27] was used to measure SD. The tool has fourteen items to assess the existence of SD.

**Couple relationship satisfaction.** Relationship satisfaction index (CSI) adapted from a previous study was applied to assess participant's satisfaction in their couple relationship [28].

The tool has six items and each item has five-point Likert scale measurements and the item ranges from "Low satisfaction" to "High satisfaction".

**Stress.** The daily stressful event measurement scale (DSEMS) adapted from a former study was used to assess daily stressful life events in the past six months [31]. This measurement scale contains ten stressful life events that the participant might experience in the past six months.

Moreover, medical history (type of diabetes, metabolic (glycemic) control, the existence of diabetic complications, the medication regimen that the patient was taking, duration of diagnosis, etc.), comorbid illness, and medication-related data were taken from the patients' medical recordings.

### Data quality assurance, data processing, and analysis

The English version of the instrument was translated to the Amharic language and retranslated back to English language to check the consistency and the Amharic version of the structured questionnaire was used for the data collection. Prior to the data collection, training and brief orientation were delivered for the data collectors and supervisors. Six male BSc nurses and three male MSc nurses were assigned as data collectors and supervisors, respectively. The data were collected in a separate room to keep the privacy of the study participant.

The collected data were checked for consistency, coding error, completeness, accuracy, clarity, and missing values before it was entered into Epi-data version 4.6. The data were further exported to SPSS version 21 for recoding, cleaning, and analyses. All continuous independent variables were categorized.

The wealth status of the participants was analyzed through the principal component analysis (PCA). All categorical and continuous variables were categorized to be between '0' and '1'. All statistical assumptions of factor analysis were checked. In addition, variables having communality value and Eigenvalues of greater than 0.5 and 1, respectively were included in the factor analysis, and the analysis was done repeatedly until all variables meet the inclusion criteria for factor analysis. Next, all eligible factor scores were computed using the regression-based method to generate one variable, wealth status. Then, the loading factors were sorted in their ascending order and they were corrected to be between four and negative four. Following this, the final scores were ranked to five quantiles as first, second, third, fourth, and fifth. Finally, ranks were coded as richest, rich, middle, poorer, and poorest, respectively.

The outcome variable was dichotomized and coded as '0' and '1', representing those who have no and have SD, respectively. Further, for continuous variables age, for instance, the Shapiro-Wilk statistic and Kolmogorov-Smirnov was used to determine which measure of central tendency is appropriate to use. Descriptive statistics like frequency, percentage, and measure of central tendency with their corresponding measure of dispersion were used to describe demographic and other variables. Tables', graphs, and texts were used to present the findings.

Furthermore, the binary logistic regression analysis was applied to identify factors associated with SD. Those variables with a p-value $\leq 0.2$ in the bi-variable analysis were entered into the multivariable logistic regression model to control the possible effects of confounder/s and to identify the significant factors. According to the Hosmer and Lemeshow test, the model was found to be adequate. Likewise, prior to identifying the significant factors, the presence of multicollinearity problem was examined using the Variance Inflation Factor (VIF), and no variable was found to have that problem. Finally, the variables which had independent correlations with SD were identified on the basis of the Adjusted Odds Ratio (AOR) and p-value with its corresponding 95% CI. Variables having a p-value $\leq 0.05$ were claimed as statistically

significant and the direction, as well as the strength of the association, was interpreted using the AOR.

### Ethical consideration

Ethical clearance was obtained from the ethical review board of the University of Gondar, and a support letter was taken from the department of Reproductive Health to be given for each respective hospital. Since the study never used any invasive procedure and biological samples of respondents, oral consent was preferred over written consent. The consent was taken after participants were informed about the risk, benefit, and their right to withdraw from the study at any time during the interview process. After reading the information to the participants, they were requested to give consent to involve in the study, and their response was written on the consent form as "agreed" provided that they had agreed to participate. Moreover, all information taken from the respondents had kept confidential and the entire data collected was only used for the purpose of this study.

## Results

### Socio-demographic characteristics of respondent

A total of 416 participants were enrolled study making a response rate of 90.04%. The mean (±standard deviation (SD)) age of the respondent was 47.8(±15.16) years. The majority (89.4%) of the respondents were Orthodox Christian followers. About two-thirds (64.7%) of the respondent had lived in an urban area and the married respondents accounted for the largest (85.3%) proportion. Moreover, slightly more than a quarter (26.4%) and a third (35.8%) of participants had attained secondary education and were private employees, respectively (Table 1).

### Medical and comorbidity characteristics

The mean (SD) duration of patients that they lived with DM was 8.22 (±5.65). Besides, higher than half (59%) of the participants were diagnosed with diabetes for more than five years. More than half (51.2%) of the participants were patients with type II diabetes and about a third (32%) of respondents had at least one diabetic complication. Moreover, 50.2% of participants had at least one comorbid illness and hypertension was the most frequent (36.6%) comorbid illness followed by hyperlipidemia (15.6%). In addition, about a third (31.2%) of hypertensive patients takes β-blockers and 16% of the patients were on diuretics. A tiny proportion (7%) of participants was taking antidepressant or other psychiatric drugs. Further, about one from every ten (9.4%) of participants had benign prostatic hyperplasia and (6.1%) of them had undergone pelvic surgery.

### Psychosocial and life style characteristics

About 59.7% of participants were found to be alcoholic. Pertaining to the nutritional status, the majority (90.9%) of respondents had BMI that falls in the normal range. More than half (59.6%) of the participants had experienced at least one stressful life event in the past 6 months of the survey. Regarding couples satisfaction, by far the largest (91.8%) proportion of the participant had satisfied with their couple relationship.

### Prevalence of SD

The prevalence of SD was found to be 69.5% (95% CI: 65.1%-73.3%). The prevalence of SD in Felege Hiwote referral hospital, Debre Markos referral hospital, and Debre Tabor general

**Table 1. Socio-demographic characteristics of men with diabetes at the three hospitals of Northwest Amhara region, Ethiopia from February 20–April 15, 2020 (n = 416).**

| Variable | Frequency (n) | Percent (%) |
|---|---|---|
| Age | | |
| <40 | 141 | 33.8 |
| 40–50 | 88 | 21.2 |
| >50 | 187 | 45 |
| Religion | | |
| Orthodox | 372 | 89.4 |
| Muslim | 38 | 9.1 |
| Protestant | 5 | 1.2 |
| Catholic | 1 | 0.2 |
| Relationship status | | |
| Single | 40 | 9.6 |
| Married | 355 | 85.3 |
| Divorced | 9 | 2.2 |
| Widowed | 12 | 2.9 |
| Educational status | | |
| Can't read and write | 83 | 20 |
| Grade 1–8 | 93 | 22.4 |
| Grade 8–12 | 110 | 26.4 |
| Diploma | 23 | 5.5 |
| Degree & above | 107 | 25.7 |
| Occupation | | |
| Government employee | 99 | 23.8 |
| Private work | 149 | 35.8 |
| Farmer | 106 | 25.5 |
| Student | 17 | 4.1 |
| Job seeker | 7 | 1.7 |
| Retired | 38 | 9.1 |
| Wealth quantile | | |
| Poorest | 86 | 20.8 |
| Poor | 109 | 26.2 |
| Middle | 107 | 25.7 |
| Rich | 68 | 16.2 |
| Richest | 46 | 11.1 |

hospital was 68.1%, 65.8%, and 73.9%, respectively. About half (53.3%) and 86.2% of type I and type II diabetes victims had experienced SD, respectively.

## Magnitude of SD in each domain

Almost all (99.5%) participants found to have the orgasmic disorder (ejaculatory problem). Similarly, participants with an arousal problem (ED) were 99.3%. Sexual pain disorder (painful orgasm and ejaculatory pain) was the SD domain which shows the lowest prevalence 42% (95%CI: 39%-45%). Further, close to half (49%) of participants were suffering from more than one form of SD (Fig 1).

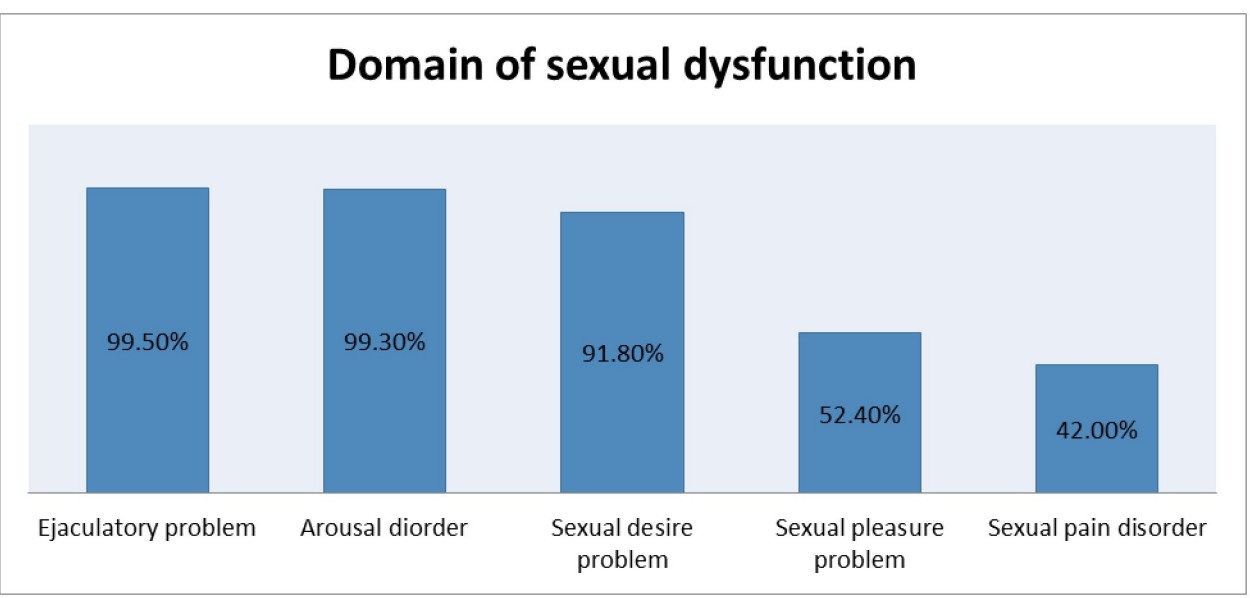

**Fig 1. Domains of sexual dysfunction among men with diabetes at the three hospitals of Northwest Amhara region, Ethiopia from February 20–April 15, 2020.**

### Factors associated with SD

Variables such as older age, rural residence, type of DM, physical inactivity, long duration of diagnosis with DM, the existence of diabetic complication, having comorbid illness, poor metabolic control, having daily stressful event, and being un satisfied with relationship were found with a p-value <0.2 in the bi-variable analysis.

In multivariable analysis variables such as older age, long duration of diagnosed with DM, physical activity, poor metabolic control, existence of diabetic complication, having comorbid illness, and relationship satisfaction have shown an independent association with SD. The odds of SD was increased by 9.6 (AOR = 9.6, 95%CI, 3.6–25.46), and 8.7(AOR = 8.7, 95%CI, 3.3–23.1) times, among participants aged between 40 and 50 and those greater than 50, respectively, than participants younger than 40 years. In addition, the likelihood of developing SD among physically active participants was reduced by 59% (AOR = 0.41, 95% CI, 0.12–0.79) than those who were physically inactive.

Participants who have been diagnosed with diabetes for more than 5 years had 10.8 (AOR = 10.8 95% CI, 5.33–21.88) times higher chance of developing SD than participants who have been diagnosed with diabetes for less than five years. The likelihood of developing SD among men with poor metabolic control rises by more than threefold (AOR = 3.57, 95% CI, 1.81–7.05) than patients with good metabolic control. Having at least one diabetic complication increased the risk of SD by three times (AOR = 3.01, 95%CI, 1.31–6.92). Likewise, the odds of SD among participants who have at least one comorbid illness were 5.07(AOR = 5.07, 95%CI, 2.16–11.9) times higher over participants who were free from comorbid illnesses.

Moreover, participants satisfied with their relationship were 85% (AOR = 0.15, 95%CI, 0.03–0.704) less likely to have SD than respondents who were unsatisfied with their relationship (Table 2).

**Table 2. Factors associated with SD among men patients with diabetes at the three hospitals of Northwest Amhara region, Ethiopia from February 20–April 15, 2020 (n = 416).**

| Variable | Sexual dysfunction | | Odds ratio (95% CI) | |
|---|---|---|---|---|
| | Yes | No | Crude(COR) | Adjusted(AOR) |
| Age | | | | |
| <40 | 60 | 81 | 1 | 1 |
| 40–50 | 50 | 38 | 1.8(1.03–3.63) | 9.6(3.6–25.50)** |
| >50 | 179 | 8 | 30.2(13.0–46.6) | 8.7(3.3–23.10)** |
| Resident | | | | |
| Rural | 93 | 54 | 1 | 1 |
| Urban | 196 | 73 | 1.9(1.01–5.40) | 1.7(0.71–4.15) |
| Type of DM | | | | |
| Type I | 114 | 99 | 1 | 1 |
| Type II | 175 | 28 | 5.43(3.35–8.78) | 0.63(0.27–1.45) |
| Physical activity | | | | |
| No | 100 | 5 | 1 | 1 |
| Yes | 189 | 122 | 0.77(0.03–0.2) | 0.41(0.12–0.7)* |
| Comorbid illnesses | | | | |
| No | 93 | 114 | 1 | 1 |
| Yes | 196 | 13 | 18.48(9.9–34.5) | 5.07(2.16–11.9)** |
| Duration of the illness | | | | |
| <5 years | 65 | 104 | 1 | 1 |
| ≥5 years | 224 | 23 | 15.58(9.18–62.76) | 10.8(5.33–21.88)*** |
| Metabolic control | | | | |
| <130 mg/dl | 39 | 57 | 1 | 1 |
| ≥130 mg/dl | 250 | 70 | 5.22(3.21–8.49) | 3.07(1.62–5.53)** |
| Daily stressful event | | | | |
| No | 100 | 67 | 1 | 1 |
| Yes | 188 | 60 | 2.1(1.37–3.21) | 1.17(0.62–2.21) |
| Couple satisfaction index | | | | |
| Satisfied | 257 | 125 | 1 | 1 |
| Unsatisfied | 32 | 2 | 0.13(0.3–0.55) | 0.15(0.03–0.704)** |
| Existence of complications | | | | |
| No | 166 | 117 | 1 | 1 |
| Yes | 123 | 10 | 8.7(4.36–17.22) | 3.01(1.31–6.98)** |

*indicate significant at p-value <0.05 and

** (<0.01), and

*** (<0.001), COR = crude odds ratio and AOR = adjusted odds ratio

Hosmer and Lemshow goodness of fit (p-value = 0.42), Multicollinearity test (VIF) = 1.28

## Discussions

SD is the commonest reproductive health problem observed among men who are aged and living with chronic non-communicable diseases like DM. The problem might end up with relationship instability, mental health disorder, and poor reproduction unless detected and managed early. However, in Ethiopia, paternal reproductive health particularly the reproductive health challenges of men with chronic illness is the most disregarded and unrecognized issue both in research and interventions. Therefore, estimating the magnitude, recognizing the most liable individuals of SD, and identifying the associated factors may have a paramount

contribution in mitigating the problem. Accordingly, this study was designed to determine the magnitude of SD and to identify factors among diabetic men attending at the three hospitals of the northwest Amhara region.

About 69.5% (65.1%-73.3%) of diabetic individuals claimed to have SD which is in line with a study conducted in Nairobi (65.1%) [19], where erectile/arousal dysfunction was the predominant domain of SD observed in both studies [19]. The prevalence of SD in this study was higher (86.2%) among type II diabetic patients than type I (53.3%). As SD and type II diabetes shared similar risk factors like aging, obesity, and high blood pressure, SD might be a common clinical entity among type II patients which could explain the observed variation [29]. Reports showed that 87.5% of individuals with type II diabetes are suffered from over-weight/obesity, which implies they are at risk of a reduced level of testosterone production that consequently result in SD [32].

This study highlighted a significant prevalence of ED (99.3%) than studies based in Israel (37%) and Nigeria (63%) [24, 33]. The tools employed across those studies were distinct; the former study conducted in Israel used the International Index of Erectile Function (IIEF), unlike the current study that applied change in the sexual functioning questionnaire, a tool which was purposely developed to assess illness, or medication-related SD [24, 33]. Likewise, the prevalence of arousal/ ED in this study was higher than studies done elsewhere in Ethiopia (69.9–85.5%) which could also be related to the abovementioned reason [20, 21].

Similar to the previous study conducted in Nairobi, the current study witnessed that the odds of SD among participants older than 50 years was 8.7 times higher than those participants younger than 40 years [19]. As a matter of fact, the level of testosterone in men declines with age at a rate of 1–2% per year starting from age 40 and older [14]. Among diabetic patients indeed, as age increases, the risk of developing peripheral neuropathy, hypertension, and impotency would also elevate, which might be the reason for an increased odds of SD [26].

Regular physical activity reduced the risk of developing SD and the current study also evidenced the same. Having regular exercise cuts the likelihood of developing SD by 59%, which is in agreement with a study conducted previously [10]. It is utterly known that physical activity enhances blood flow to the genitalia and promotes sexual desire. Similarly, it has a favorable effect on testosterone production, a hormone that promotes sexual desire and behavior [32]. Thus, promoting physical activity would strengthen the sexual performance of individuals apart from preventing other chronic illnesses that have an adverse impact on the sexual health of men.

Consistent with a study conducted earlier, this study indicates that nutritional status doesn't show any association with SD [34]. Patients, who lived with diabetes for greater than five years, were 10.5 times more at risk of developing SD than their counterpart. This finding is analogous with other studies conducted before [25, 35]. Since the risk of developing micro and macrovascular diabetic complications become higher when the duration of living with the illness increases, the risk of developing SD might also be elevated [25]. Similarly, although the drugs used for the management of diabetes don't have a direct relationship with SD, taking those drugs for a longer period would increase the risk of heart failure and weight gain that have a deleterious impact on men's sexual function [18].

The likelihood of developing SD among individuals with diabetic complications like retinopathy, nephropathy, and neuropathy was three times higher than patients who were free from diabetic complications, which is supported by a former study [25]. People with neuropathy obviously have poor penile innervation that interferes with the normal dilation of penile blood vessels and compromises the relaxation of penile muscles for erection that eventually affects sexual function [25, 26].

Having other comorbid illnesses including HTN, cardiovascular disease, hyperlipidemia, and others raise the odds of SD by 5.07 times. The finding was congruent with another study that shows patients with other concomitant medical conditions increase the risk of developing SD [24, 36, 37]. This might be due to: (i) different comorbid illnesses could solely alter the sexual function of individuals, for instance, renal disease results in significant endocrine disturbances, including hypogonadism due to reduced renal clearance; (ii) the drugs used to manage those comorbid illnesses have a proven side effect on sexual function, for instance, antihypertensive drugs, reduce blood flow to the reproductive organs of men that ultimately affects the penile erection capacity; and (iii) the psychological impact of having chronic illness would further compromise the sexual desire of individuals [6, 36, 38]. This study suggested that comorbidity interferes with the reproductive health of individuals on top of its challenge to stabilize patient's blood glucose levels, putting them at higher risk of death. Therefore, it should be noted that preventing concomitant illnesses, managing and controlling them at the earlier stage would help to maintain an individual's reproductive health and life.

Similarly, patients with poor metabolic control were at a greater risk of developing SD than those with good metabolic control. Another study also witnesses the association in that the odds of developing SD was higher among respondents with poor metabolic control than their counterparts [25, 29]. Poor metabolic control is associated with an increased risk of long-term macro and micro-vascular complications that might have a greater impact on the occurrence of SD [1, 29]. In other words, the severity of the illness (DM) is presumed to be raised among patients with poor metabolic control that concomitantly increase the risk of diabetic complications, including SD.

Furthermore, the study revealed that participants who were satisfied with their couple relationship were 85% less likely to develop SD than those who were unsatisfied, which is similar to a previous finding [31]. Since sexual function is the cumulative effect of the vascular, neurologic, hormonal, and psychologic system, couple un satisfaction have an impact on the psychological well-being of an individual that might reduce sexual desire [31]. The linkage of SD and relationship satisfaction is interplayed; to put it simply, SD weakens the bond between couples as a result of poor sexual satisfaction, and unhealthy relationship results in poor sexual desire and performance. It would be possible to infer that having a good relationship is crucial not only to maintain the mental, emotional, and physical health dimensions but also improves the reproductive health of individuals. Thus, people, in particular living with diabetes are advised to establish a healthy relationships. Healthcare professionals shall promote strategies in maintaining healthy relationships with their clients. In nutshell, given the devastating reproductive, mental, psychological, and emotional health impact of SD, the Amhara regional health office and the federal government should work jointly to tackle SD and its contributing factors [39, 40].

## Strength and limitation of the study

The study reported the burden of SD that would help to expand individual's sphere of knowledge in the field. In addition, it was conducted in health institutions that is presumed to be helpful to acquire reliable clinical data. Although all possible strategies have been applied to reduce bias like recruiting male interviewers and underway the interview in the most private room, the study might still have introduced social desirability bias due to the sensitivity of the research question and the nature of the data collection technique (face-to-face interviewer-administered questionnaire). As there are factors which didn't include, the study might have exposed to confounding effect. Further, unable to measure the testosterone level of the participants could be another limitation of the study to show the effect of diabetes on men's sexuality.

## Conclusions

The study remarks that more than two-thirds of men with diabetes have experienced SD. Older age, living longer duration with the illness, poor metabolic control, lack of physical activity, having diabetic complications, experiencing comorbid illness, and being unsatisfied with couple relationship were factors contributed to developing SD. Special emphasis should be given to older patients and those who have been living with diabetes for a longer time. Moreover, participants should be promoted to engage in regular physical activity and other healthy practices to maintain good glycemic control so as to prevent different diabetic complications, including SD. Marriage counseling could be another strategy to mitigate SD. The finding is an alarming issue, demands strengthening chronic care by promoting personal and behavioral change communications.

## Supporting information

**S1 File.**
(SAV)

## Acknowledgments

We would be really delighted to express our appreciation for the study participants for providing us the basic information for our research.

## Author Contributions

**Conceptualization:** Eskedar Getie Mekonnen, Hedija Yenus Yeshita, Alehegn Bishaw Geremew.

**Formal analysis:** Eskedar Getie Mekonnen.

**Investigation:** Eskedar Getie Mekonnen.

**Methodology:** Eskedar Getie Mekonnen, Hedija Yenus Yeshita, Alehegn Bishaw Geremew.

**Supervision:** Hedija Yenus Yeshita, Alehegn Bishaw Geremew.

**Writing – original draft:** Eskedar Getie Mekonnen.

**Writing – review & editing:** Eskedar Getie Mekonnen.

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
