## [Decision Letter · Decision Letter 0]

26 Apr 2021

PONE-D-20-32177

Sexual dysfunction among men with diabetes mellitus attending chronic out-patient department at the three Hospitals of northwest Amhara region, Ethiopia: Prevalence and associated factors.

PLOS ONE

Dear Dr. Getie,

Thank you for submitting your manuscript to PLOS ONE. After careful consideration, we feel that it has merit but does not fully meet PLOS ONE’s publication criteria as it currently stands. Therefore, we invite you to submit a revised version of the manuscript that addresses the points raised during the review process.

Both reviewers raised some concerns with the study methodology and the grammar/presentation of the manuscript. The reviewers' comments can be viewed in full, below.

We look forward to receiving your revised manuscript.

Kind regards,

Natasha McDonald, PhD

Associate Editor

PLOS ONE

Journal Requirements:

2. In the Methods, please clarify that participants provided oral consent. Please also state in the Methods:

- Why written consent could not be obtained

- Whether the Institutional Review Board (IRB) approved use of oral consent

- How oral consent was documented

For more information, please see our guidelines for human subjects research: https://journals.plos.org/plosone/s/submission-guidelines#loc-human-subjects-research

4. Please ensure you have thoroughly discussed any additional potential limitations of this study within the Discussion section, including the potential impact of confounding factors.

5. We suggest you thoroughly copyedit your manuscript for language usage, spelling, and grammar. If you do not know anyone who can help you do this, you may wish to consider employing a professional scientific editing service.  

Reviewers' comments:

Reviewer's Responses to Questions

**Comments to the Author**

1. Is the manuscript technically sound, and do the data support the conclusions?

Reviewer #1: Partly

Reviewer #2: Yes

2. Has the statistical analysis been performed appropriately and rigorously? 

Reviewer #1: Yes

Reviewer #2: Yes

3. Have the authors made all data underlying the findings in their manuscript fully available?

Reviewer #1: Yes

Reviewer #2: No

4. Is the manuscript presented in an intelligible fashion and written in standard English?

Reviewer #1: No

Reviewer #2: Yes

5. Review Comments to the Author

Reviewer #1: The present study investigated the prevalence and the associated risk factors for sexual dysfunction in a popluation of men from Ethiopia. The topic is interesting and deserves attention. However, the manuscript is seriously flawed by the poor quality of the English used.

I suggest an extensive revision of the English by a specialist to correct the enormous number of mistakes in grammar and typos.

The last three sentences of the introduction shoud be move to the discussion

Details on the CSFQ should be transferred in a supplementary files to shorten the methods section

In the methods, definition of the microvascular complications of diabetes should be given (retinopathy, nephropathy, neuropathy)

Do the authors measure the testosterone levels of the participants in the study? Have hypogonadic men been excluded? Have the inteference of drugs interfering with sexual function (i.e. beta-blockers, anti-depressants, etc) been considered?

Reviewer #2: This cross-sectional study was conducted among 462 diabetic men in Ethiopia, and sexual dysfunction was found in 69.5% based on questionaires collected. No control group was included, which should be underlined as a major limitation. However, due to the high number of patients included, the manuscript deserves publication after revision.

Abstract - Results, line 2: Should "disproportionately" be replaced by "proportionately"?

Abstract - Conclusion, line 2: Should "physical activity" be replaced by "lack in physical activity"?

Main text - Background, line 28: "hypertension increased the problem". Are you sure it is the hypertension itself? Could erectile dysfunction rather be caused by the anti-hypertension treatment according to e.g. Fedder et al., 2013?

Reference list: For at least 10 references, the name of the journal is not mentioned. It should be included.

6. PLOS authors have the option to publish the peer review history of their article (what does this mean?). If published, this will include your full peer review and any attached files.

Reviewer #1: **Yes: **Maria Ida Maiorino

Reviewer #2: No

---

## [Author Response · Author response to Decision Letter 0]

4 Jun 2021

May 18/2021

Point-by-point response 

Dear the editor and reviewers, we found your comments to be crucial for enhancing our scholarly work. We are really grateful enough to express our appreciation for your comments. Appreciating your effort and valuable comments, we have provided possible reflections on the raised concerns and questions. Kindly find our response hereunder. 

A. Editor’s comment 

Authors’ response: comment has been accepted.

2. In the Methods, please clarify that participants provided oral consent. Please also state in the Methods: why written consent could not be obtained? Whether the Institutional Review Board (IRB) approved use of oral consent?, and how oral consent was documented?

Authors’ response: Dear, the reason why we could not obtain written consent was as the study never used biological samples from the participants and applied any invasive procedure, we have taken oral informed consent that approved by the Institutional review board committee of the University of Gondar. 

As per your recommendation, the ethical approval statement has appeared only in the method section of the manuscript and the detail of ethical consideration was presented in the updated version. 

Author’s response: Dear, we had collected the data for the outcome variable using the Changes in Sexual Functioning Questionnaire (CSFQ-14) adapted from the changes in sexual functioning questionnaire short form (CSFQ14) that any scholar can access it online. Similarly, for the independent variables, daily stressful event measurement scale (DSEMS) and relationship satisfaction (CSI), for instance, were taken from the previously published articles that were cited in the main document. We can supply a copy of both the Amharic (local language) and English language tool that we used to collect the data if that is necessary.

4. Please ensure you have thoroughly discussed any additional potential limitations of this study within the discussion section, including the potential impact of confounding factors.

Author’s response: comment accepted and the potential limitation of the study has been added in the newly revised manuscript. 

5. We suggest you thoroughly copyedit your manuscript for language usage, spelling, and grammar. If you do not know anyone who can help you do this, you may wish to consider employing a professional scientific editing service. 

Author’s response: comment accepted and the language usage has been improved.

Author’s response: comment accepted and the necessary modification has been undertaken. 

B. Reviewer #1

1. The manuscript is seriously flawed by the poor quality of the English used.

I suggest an extensive revision of the English by a specialist to correct the enormous number of mistakes in grammar and typos.

Authors’ response: comment accepted and necessary modification has been done.

2. The last three sentences of the introduction should be move to the discussion

Author’s comment: comment accepted and the statements have been taken to the discussion section. 

3. Details on the CSFQ should be transferred in supplementary files to shorten the methods section.

Author’s comment: we appreciate your concern dear, and we have precisely presented it in the methods section.

4. In the methods, definition of the microvascular complications of diabetes should be given (retinopathy, nephropathy, neuropathy)

Authors’ comment: comment accepted and it has been modified accordingly.

5. Do the authors measure the testosterone levels of the participants in the study? Have hypogonadic men been excluded?

Authors’ comment: Dear, the higher burden of hypogonadism among type II diabetic patients was previously evidenced, which could be one of a pathological pathway for the development of SD. We believe that excluding hypogonadic men in our study would largely affect our study because of introducing bias, in particular, selection bias. Patients with SD might obviously have hypogonadism and excluding these people would severely affect the outcome variable and lead to miss inference. Although measuring the level of testosterone is truly important, we didn’t measure it as the test was not available in the study setting and even in our country. 

6. Have the interference of drugs interfering with sexual function (i.e. beta-blockers, anti-depressants, etc.) been considered?

Authors’ comment: the existence of other comorbidities along with their medication, as well as other drugs with a possible side effect of SD was examined. The data were collected through reviewing the medical recording of participants as those factors might have confounding effect. Then the medication history was considered to include in the analysis but it doesn’t fulfil the chi-square assumption. We have included the descriptive data on the currently updated document. 

C. Reviewer #2

1. This cross-sectional study was conducted among 462 diabetic men in Ethiopia, and sexual dysfunction was found in 69.5% based on questionnaires collected. No control group was included, which should be underlined as a major limitation. However, due to the high number of patients included, the manuscript deserves publication after revision.

Authors’ comment: Dear, as our primary objective was to investigate the prevalence of SD among men with diabetes, we didn’t consider a control group. Dear, if we had established a control (patient without SD) and case group (participants with SD), the result would have differed from our primary objective (prevalence of SD) and we wouldn’t have estimated the prevalence. Thus, we don’t believe that not considering the control group wouldn’t be the possible limitation of the study. However, we are ready to accept your comment if you are not satisfied with the feedback given. 

2. Abstract - Results, line 2: Should "disproportionately" be replaced by "proportionately"?

Author’s comment: comment accepted and corrected accordingly. 

3. Abstract - Conclusion, line 2: Should "physical activity" be replaced by "lack in physical activity"?

Authors’ comment: comment accepted and modified accordingly

4. Main text - Background, line 28: "hypertension increased the problem". Are you sure it is the hypertension itself? Could erectile dysfunction rather be caused by the anti-hypertension treatment according to e.g. Fedder et al., 2013?

Authors comment: SD among hypertensive patients are multifactorial. Not only antihypertensive medications have a deleterious effect on an individual’s sexual function rather hypertension by itself have an impact on patients sexual function associated with its effect on the blood vessels of the genitalia. Likewise, as you said, it’s undeniable evidence that anti-hypertensive agents like β-blockers and diuretics could have the potential to further impair an individual’s sexual function through reducing blood flow to the reproductive organs. Moreover, the psychological impact of chronic illnesses are another contributing factor for developing SD. All in all, you are right both factors (hypertension and the drugs) are responsible and they have been included in the study and examined for its association with SD. 

5. Reference list: For at least 10 references, the name of the journal is not mentioned. It should be included.

Authors’ comment: comment accepted and the names of the journals have been included except for the gray literature.

---

## [Decision Letter · Decision Letter 1]

22 Jul 2021

PONE-D-20-32177R1

Sexual dysfunction among men with diabetes mellitus attending chronic out-patient department at the three Hospitals of northwest Amhara region, Ethiopia: Prevalence and associated factors.

PLOS ONE

Dear Dr. Getie,

Thank you for submitting your manuscript to PLOS ONE. After careful consideration, we feel that it has merit but does not fully meet PLOS ONE’s publication criteria as it currently stands. Therefore, we invite you to submit a revised version of the manuscript that addresses the points raised during the review process.

We look forward to receiving your revised manuscript.

Kind regards,

Ishag Adam, MD, PhD

Academic Editor

PLOS ONE

Journal Requirements:

Reviewers' comments:

Reviewer's Responses to Questions

**Comments to the Author**

1. If the authors have adequately addressed your comments raised in a previous round of review and you feel that this manuscript is now acceptable for publication, you may indicate that here to bypass the “Comments to the Author” section, enter your conflict of interest statement in the “Confidential to Editor” section, and submit your "Accept" recommendation.

Reviewer #1: All comments have been addressed

Reviewer #2: All comments have been addressed

2. Is the manuscript technically sound, and do the data support the conclusions?

Reviewer #1: Yes

Reviewer #2: Yes

3. Has the statistical analysis been performed appropriately and rigorously? 

Reviewer #1: Yes

Reviewer #2: Yes

4. Have the authors made all data underlying the findings in their manuscript fully available?

Reviewer #1: Yes

Reviewer #2: Yes

5. Is the manuscript presented in an intelligible fashion and written in standard English?

Reviewer #1: Yes

Reviewer #2: Yes

6. Review Comments to the Author

Reviewer #1: I have no further comment. The raised issues have been addressed and the quality of manuscript has improved.

Reviewer #2: I have no further comments. I think it is a nice paper, which deserves publication. Hope you will continue to work in this field.

7. PLOS authors have the option to publish the peer review history of their article (what does this mean?). If published, this will include your full peer review and any attached files.

Reviewer #1: No

Reviewer #2: No

---

## [Author Response · Author response to Decision Letter 1]

24 Jul 2021

As there were no concerns and questions raised by both the editor and reviewers, responses are not applicable.

---

## [Editor Report · Decision Letter 2]

29 Jul 2021

Sexual dysfunction among men with diabetes mellitus attending chronic out-patient department at the three Hospitals of northwest Amhara region, Ethiopia: Prevalence and associated factors.

PONE-D-20-32177R2

Dear Dr. Getie,

We’re pleased to inform you that your manuscript has been judged scientifically suitable for publication and will be formally accepted for publication once it meets all outstanding technical requirements.

Kind regards,

Ishag Adam, MD, PhD

Academic Editor

PLOS ONE
---

## [Editor Report · Acceptance letter]

2 Aug 2021

PONE-D-20-32177R2 

Sexual dysfunction among men with diabetes mellitus attending chronic out-patient department at the three Hospitals of northwest Amhara region, Ethiopia: Prevalence and associated factors. 

Dear Dr. Getie Mekonnen:

I'm pleased to inform you that your manuscript has been deemed suitable for publication in PLOS ONE. Congratulations! Your manuscript is now with our production department. 

Kind regards, 

on behalf of

Professor Ishag Adam 

Academic Editor

PLOS ONE